# Collembola of the Cavalum and Landeiro Caves (Madeira, Portugal) [note 1]

**DOI:** 10.3390/insects14060525

**Published:** 2023-06-05

**Authors:** Enrique Baquero, Javier I. Arbea, Élvio Nunes, Dora Aguin-Pombo, Eduardo Mateos, Rafael Jordana

**Affiliations:** 1Department of Environmental Biology, Faculty of Sciences, University of Navarra, 31080 Pamplona, Spain; rjordana@unav.es; 2Institute for Biodiversity and Environment BIOMA, University of Navarra, Irunlarrea 1, 31008 Pamplona, Spain; 3c/Ría de Solía 3, ch. 39, 39610 El Astillero, Spain; jarbeapo@gmail.com; 4Faculdade de Ciências da Vida, University of Madeira, 9000-390 Funchal, Portugal; elviosn@gmail.com (É.N.); aguin@staff.uma.pt (D.A.-P.); 5Centro de Investigação em Biodiversidade e Recursos Genéticos (CIBIO), 4485-661 Vairão, Portugal; 6Departament de Biologia Evolutiva, Ecologia i Ciències Ambientals, Facultat de Biologia, Universitat de Barcelona, Avinguda Diagonal 643, 08028 Barcelona, Spain; emateos@ub.edu; 7Institut de Recerca de la Biodiversitat (IRBio), Universitat de Barcelona, 08007 Barcelona, Spain

**Keywords:** springtails, new species, volcanic cavities, cave fauna

## Abstract

**Simple Summary:**

The cave fauna of the Macaronesian archipelagos is rich in endemic species. In this region, most studies have been carried out in the Azores and the Canary Islands. In contrast to these archipelagos, the arthropod fauna of the lava tubes of the Madeira has not been well studied. Madeira is the only island in this archipelago with lava tube complexes suitable for endogenous arthropod species. In the two known complexes, San Vicente and Machico (Cavalum), 13 endemic species have been recorded; among these, there are two critically endangered cave spiders of the genus *Centromerus*. These caves not only have no protection measures, and while they are not exploited for tourism, they are under serious threats by the frequent passage of people. The San Vicente complex, the largest on the island, has undergone profound structural modifications for its tourist exploitation but still, since its inauguration in 1996, receives thousands of visitors per year. Until now, faunal studies of Madeiran caves have been the result of casual collecting. This work is the first monitoring study of Madeira’s cave fauna. One-year monitoring has been done in four lava tubes of Machico: the Cavalum lava tubes (I, II, III) and the Landeiros tube. This study aimed to make a species checklist of the cave fauna present. So far, only three species of Collembola have been described for the Madeira lava tubes. Here we describe four new species, *Neelus serratus* Jordana & Baquero sp. nov., *Coecobrya decemsetosa* Jordana & Baquero sp. nov., *Coecobrya octoseta* Jordana & Baquero sp. nov., and *Sinella duodecimoculata* Jordana & Baquero sp. nov., and we update the checklist of the Collembola of lava tubes from three to 16 species. At present, the Machico complex caves represent the island’s only natural lava tube habitats and are probably the main contributor to Madeira’s cave biodiversity.

**Abstract:**

The cave fauna of the Macaronesian archipelagos is rich in endemic species. Unlike the Azores and the Canary Islands, the cave fauna of the Madeira archipelago is little known. The only two cave complexes studied, Machico and São Vicente, lack protection measures. São Vicente is seriously threatened because it is being exploited for touristic purposes, while the Machico complex, the only one in its natural state, is open to the public but without any control. The importance of the conservation of this cave fauna is indisputable. So far, of the 13 cavernicolous species recorded, two of them—genus *Centromerus*—are critically endangered. Apart from occasional sampling, no monitoring study has ever been carried out. The aim of this work has been to make a species checklist of the cave fauna of the Machico complex, the least studied so far. For this purpose, during 2001–2002, a monitoring study was carried out using traps and manual collections in the lava tubes of Landeiros and Cavalum (I, II, III). Fourteen species of springtails were recorded. Of these, four are new species, *Neelus serratus* Jordana & Baquero sp. nov., *Coecobrya decemsetosa* Jordana & Baquero sp. nov., *Coecobrya octoseta* Jordana & Baquero sp. nov., and *Sinella duodecimoculata* Jordana & Baquero sp. nov., and one is a new record for the archipelago, *Lepidocyrtus curvicollis* Bourlet, 1839.

## 1. Introduction

Macaronesia is a hotspot of hypogea faunal diversity. Most cave fauna studies in this region have mainly focused on the Canary Islands and, to a lesser extent, the Azores. Comprehensive speleological studies carried out by local biologists over the last 40 years in these archipelagos have led to the discovery of rich cavernicolous fauna [1,2]. Likewise, studies of other non-cave subterranean environments, such as the Mesovoid Shallow Substratum (MSS) and the pyroclastic deposits, have provided further data for endogenous biodiversity [3,4]. At least 238 troglobiont species occur in Canarian lava tubes, 34% of which are found in Tenerife [1]. Biogeographically, the Macaronesia fauna is not only very rich but also shows interesting features. Many species are endemic to a single island or lava tube; others show a high degree of adaptation to cave life; some are relicts without representatives in epigean ecosystems; while others belong to lineages that have undergone extensive radiation [1,4]. Cockroaches of the genus *Loboptera* [5,6], *Dysdera* spiders [7], *Dolichoiulus* millipedes [8], *Cixius* planthoppers [9], or carabids of the genus *Trechus* [10] are some of the most notable cases of species radiations.

Unlike the Canary Islands and the Azores, the arthropod fauna of the Madeira has not been well studied. Of the five islands of this archipelago, Madeira is the only one with caves of suitable dimensions for endogean fauna to prospect. More than a dozen caves are known, of which nine have been studied geologically [11]. The mapping of the island’s volcanic cavities is still incomplete; therefore, this number cannot be considered definitive. The steep and rugged relief of the island and the lush vegetation that covers it make this work difficult. Indeed, erosion-induced landslides have made many of them no longer available for exploration. Speleological studies on Madeira’s cave-dwelling fauna have focused on two basaltic lava tube complexes: the San Vicente and Machico complexes [12]. The former includes the Cardais lava tubes. These are the largest in Madeira with 1100 m in length. They consisted of a complex of four interconnected sub-parallel tubes resulting from the bifurcation of a main channel. The Mio-Miocenic basaltic complex of Machico [13] includes the Cavalum lava tubes, a net of four tubes (I, II, II, IV) from 50 to 300 m in length. Nearby is the 85 m long Landeiros cave with two main branches [11].

The first three speleological works from Madeira date from the 1950s and early 1960s. Vandel (1960) [14] also described the first Madeiran troglobitic species, the woodlouse *Trichoniscus bassoti* Vandel, 1960 [14], which is apparently the only Macaronesian troglobitic species living on two archipelagos, Madeira and the Canary Islands. Between the 1980s and 1990s, after a twenty-year publication gap, 20 additional studies came to light. Since the description of a new cave-dwelling pseudoscorpion from the Cavalum caves, *Paraliochthonius cavalensis* Zaragoza, 2004 [15], and more recently in 2014, of two millipedes of the genus *Cylindroiulus*, one from Cardais and the other from Cavalum complex [16], no further work has been published. In total, 35 species from 19 families of arthropods are known from Madeira lava tubes. Of these, 14 are endemic and eight are strictly cave-dwelling, *Eukoenenia madeira* Strinati & Condé 1995 [17], *Centromerus sexoculatus* Wunderlich, 1992 [18], *Trichoniscus bassoti* Vandel, 1960 [14], *Cylindroiulus julesvernei* Reboleira and Enghoff, 2014 [16], *Cylindroiulus oromii* Reboleira and Enghoff, 2014 [16], *Thalassophilus pieperi* Erber, 1990 [19], *Medon vicentensis* Serrano, 1993 [20], and *Paraliochthonius cavalensis* Zaragoza, 2004 [15]. Close cavernicolous species of the same genera occur in neighboring archipelagos of the Canary Islands, such as the pseudoscorpions, *Paraliochthonius martini* Mahnert, 1989 [21] and *Paraliochthonius tenebrarum* Mahnert, 1989 [21], and the carabid *Thalassophilus subterraneus* Machado, 1989 [22], or *Thalassophilus azoricus* Oromi & Borges, 1991 [23], in the Azores [24,25].

Sixteen species of arthropods were found in the Cardais lava tube complex, five of which are troglobites. In the 1990s, extensive structural modifications were undertaken in these caves to exploit the site for tourism. In 1996 they were opened to the public and in 2017 alone, the number of visitors was over 150,000 “www.grutasecentrodovulcanismosaovicente.com (accessed on 2 June 2022)”. As no further speleological studies have been undertaken since its opening to tourism, it is not known whether this high number of visitors may have created imbalances or irreparable damage to this fragile ecosystem. The Cavalum lava tubes have been listed in the Madeira Geological Heritage Inventory as a geosite of interest (MO4) since 2012 [26]. Unlike the São Vicente Caves, they have not undergone structural alterations. These caves are not currently under any kind of protection, and their entrance is open with unrestricted control. However, they are considered vulnerable and urgent measures have been proposed [26]. The Machico caves, despite their small size, have remarkable diversity. They may currently represent the largest exponent of Madeira’s cave biodiversity. Twenty species have been reported from this complex, four of which are common to both lava complexes.

Knowledge of Madeira’s cave fauna is far from complete. For some species, the name of the cave where they were collected is not known, nor even the lava tube complex where the samples were taken. For example, the only reference to the springtails from Madeiran caves are three species of the Cavalum complex *Disparrhopalites patrizii* (Cassagnau & Delamare, 1953) [27], *Onychiurus ghidinii* (Denis, 1938) [28] and *Onychiurus circulans* Gisin, 1952 [29,30,31]. However, the two papers on the Collembola fauna of caves do not indicate from which cave(s) the specimens of these three trogophilous species were taken. All faunistic studies published so far on Madeira lava tubes are the result of casual collections in the largest and best-known caves of the two complexes.

The objective of this sampling was to make a checklist of the cave-dwelling species present in four lava tubes of the Machico complex, the Cavalum caves (I, II, III) and Landeiros. This work is the result of an unpublished one-year monitoring study during the period 2001–2002 [32]. As a result of the material from this study, a new cavernicolous pseudoscorpion, *Paraliochthonius cavalensis* [15], and a species of *Cylindroiulus*, *C. oromii*, have been described. In addition, a list of the springtails identified has been provided by [33]. Here we describe four additional new species of Collembola and provide an account of the 14 species of springtails present.

## 2. Materials and Methods

### 2.1. Geomorphological Characteristics of Studied Lava Tubes

The caves of Cavalum (the term refers to demon or devil) and the Landeiros are located in the parish of Machico, in the sites of Maroços and Landeiros, respectively (Figure 1). They originated from a basaltic drain that went from a volcanic apparatus located in Santo da Serra to a paleo-valley in Machico [12]. The Caves of Cavalum are composed of four volcanic cavities close to each other, inserted in a small forest with a predominance of introduced plant species, such as *Eucalyptus* sp. and *Acacia* sp., along with some indigenous species, such as *Myrica faya* Aiton, *Laurus novocanariensis* Rivas Martínez, *Vaccinium padifolium* Sm., and *Erica scoparia* L. The cave of the Landeiros is 500 m away from these on the left bank of a brook, within a private property with an anonas plantation (*Annona cherimolia* Mill.). In this study, four caves were covered: Cavalum I, Cavalum II, Cavalum III, and Landeiros.

### 2.2. Site

Cavalum I (coordinates 32.73272, −16.78383, 140 m asl; datum WGS84) (Figure 1) reaches a total length of 300 m [34]. The temperature inside varies between 15 and 17 °C and the relative humidity between 75 to 86 % (Table 1). The entrance, which was covered by vegetation, is 2 m high and 3.5 m wide and leads us, through a corridor of 30 m, to a small hall with a clayey stratum [11]. To the northwest there is a narrow tube that gives access to the outside and in the southwest, there is a small opening that gives access to a wide chamber, with a clayey substrate with small rocky blocks resulting from ceiling detachments [34]. Progressively, this chamber continues in a downward tilt tube, which decreases in both height and width, ending in a narrow laminar passage with a mud-covered floor. In most of this passage the ceiling and walls are covered with stalactites. In the opposite direction to the chamber there is a small corridor, with a minimum height of 0.5 m, where an air stream is noticeable. After this corridor comes a new large chamber up to 10 m high, with large rocky blocks on the ground resulting from ceiling detachments. To the south of this enclosure there is a new regular conduit that extends to another more or less extended room. From here to the end of this tunnel, the ground is covered with mud, and there are even some puddles [35].

Cavalum II (coordinates 32.73252, −16.78472, 159 m asl) is a regular north-south orientated tube 110 m long [34]. Inside the maximum height is 8 m, the temperature varies between 15 to 17 °C, the relative humidity is around 85 to 95% (Table 1), and the substrate is composed of rocky blocks, some of which are large. At the end of the tube there are water infiltrations and a reduction of the ceiling. From the main tube come two small secondary laminated channels that again intersect the main tube. One of these channels is on the right side, around the middle of the passage, and it is necessary to climb the wall to access it. The other is at the end of the main corridor, on the left side, and unlike the previous one, water infiltration occurs.

Cavalum III (coordinates 32.73241, −16.78477, 156 m asl) (Figure 2) is at a quota of 170 m, only 10 m west of Cavallum II [11]. It is a single, irregular, east-west-oriented tube reaching approximately 92 m in total length with an entrance that is 5 m high by 2.5 m wide [34]. The temperature inside is around 15 to 17 °C and the relative humidity is 85 to 95% (Table 1). Halfway along the route, there is a passage that is 0.5 m high with a muddy substrate. From here to the end of the pipe, there is muddy soil noticeable along the entire length of the tube. Here the walls and ceiling are covered with lava drips and there are some deep cracks and detachments from the ceiling that make progress difficult.

The Landeiros cave (coordinates 32.73194, −16.78988, 185 m asl) is difficult to locate as it has a very discreet circular entrance of only 0.6 m in diameter and it is within a private property. The total length of the tube is approximately 85 m. The interior temperatures range from 15 to 17 °C and the relative humidity range is 85 to 95% (Table 1). Along the length of the tube, in addition to the existence of large blocks on the ground resulting from ceiling detachments, there are deep cracks in the ceiling and water infiltration, so that the danger of collapse is imminent. The end of the main tube is blocked by a collapse that makes progress impossible. However, there are two branches: one on the right, larger and steeper, which connects back to the main tube, and another to the left, which corresponds to a small well with clay deposits at the bottom.

### 2.3. Environmental Parameters

Determination of soil pH.

Samples of sandy or clay soil were collected in all the caves: CVI (four samples), CVII (1), CVIII (1), and Landeiros (1). The samples were dried in an oven at 80 °C and then passed through a 1 mm sieve. From each sample, three 10 g subsamples were chosen. To each of these, 15 mL distilled water was added. This solution was stirred for 1 min, allowed to stand for 1 h and, after this time, was again stirred and filtered. The pH was measured in the supernatant of the solution (Forster, 1995) with a Crison Micro-pH 2001 m (Crison Instruments, S.A., Barcelona, Spain). This operation was repeated three times for each sample. The pH value corresponds to the average of the three measurements. In the Landeiros cave the pH of a water sample was also measured.

In addition to pH, temperature and humidity were recorded inside the caves (Table 1). For temperature, a maximum and minimum thermometer was left inside each cave between two sampling periods, while humidity was recorded with a thermo-hygrometer.

### 2.4. Sampling

As the cave-adapted arthropods are small in size and their populations are also usually very small, in addition to hand collection it is necessary to use pitfall traps with deception, which is a very effective method to capture these animals. Cylindrical PVC containers with 4.5 cm diameter and 7 cm height were used as traps, and for each trap 40 mL of modified Turquin liquid was used, consisting of a mixture of 1000 mL of beer, 5 mL of acetic acid, 5 mL of formaldehyde, and 10 g of chloral hydrate [36]. Turquin’s liquid is an attractive but also a preservative [37,38]. However, other groups of arthropods are not attracted by Turquin bait but by food. To attract and concentrate these arthropods in the vicinity of the traps, a small piece of liver or blue cheese was suspended with a clip above the opening of the pitfall trap [37,38]. The traps were buried and surrounded with stones to create access bridges. The aim of this study was to identify the arthropod fauna present in these caves. For this purpose, we use an intensive sampling method in a short period of time. The number of traps chosen was decided according to the length of the caves and the sampling objective. In each sampling period, old traps were removed and replaced by new ones. There were always 17 traps in Cavalum I, 12 in Cavalum II, nine in Cavalum III, and six in Landeiros. The number of sampling days ranged from 28 to 65 depending on the cave (Table 1). Visually located individuals were collected with an entomological aspirator or a paintbrush. All specimens were preserved in 80% alcohol with 2% glycerin.

Sample codes indicate the trapping method: ‘C’ for hand collection and ‘Pn’ for Pitfall, where ‘n’ is the trap number (see Figure 1 for location).

### 2.5. Material Procesing

After preliminary sorting 1091 specimens were studied. Some of them were mounted in Hoyer’s medium for examination under a compound microscope using phase contrast and DIC objectives. Some were cleared previously in Nesbitt’s fluid. The remaining samples were fixed and stored in 70–80% ethyl alcohol.

The slides were observed under two microscopes: an Olympus BX51-TF (Olympus Group, Tokyo, Japan) with multiple viewing and phase contrast and a Zeiss model Axio Imager.A1 with differential interference contrast (DIC). For measurements, a U-DA drawing attachment UIS (Universal Infinity System) and a scale calibrated with a slide by Graticules Ltd., Cambridge, UK (1 mm divided in 100 parts) were used. For SEM (Scanning Electron Microscopy) the specimens were dehydrated using series of ethyl alcohol followed by critical-point drying in CO_2_, then mounted on aluminum SEM stubs, and coated in an Argon atmosphere with 16 nm gold in a sputter-coater Emitech Ltd., Strovolos, Chipre, model K550. SEM observations were made with a FE-SEM Zeiss model Sigma 300 VP (Zeiss, Oberkochen, Germany).

The specimens studied will form part of the collections of the Museum of Zoology of the University of Navarra and the Collection of Insects of the University of Madeira, although some paratypes will be deposited in the Museum of Natural Sciences of Madrid.

### 2.6. Photographing

Live specimens were taken from the caves and brought to the laboratory where they were placed in a terrarium with cave substrate. Before being photographed, the specimens were placed in a Petri dish with cave substrate and left for a few minutes in a refrigerator. Photographs were taken with an Olympus DP11 camera coupled to an Olympus SZH10 stereomicroscope. The insects were kept in the dark and only the cold light of the transilluminator was used.

### 2.7. Nomenclature

The following literature sources have been used for the nomenclature. For *Neelus*, Schneider (2017) [39]. For *Coecobrya*: labial chaetotaxy, Gisin (1967) [40]; clypeal chaetotaxy, Zhang et al. (2016) [41]; labial papilla, Fjellberg (1999) [42]; tergal chaetae groups, Chen & Christiansen (1993, 1997) [43,44]; dorsal head, Mari Mutt (1979) [45] modified by Soto-Adames (2008) [46]; head and body macrochaetae, Szeptycki (1979) [47] modified by Jordana & Baquero (2005), Zhang et al. (2011) and Jordana (2012) [48,49,50]; and trunk specialized chaetae (S-chaetae), Zhang & Deharveng (2015) [51]. For *Sinella*: labial chaetae, Gisin (1967) [40]; head chaetotaxy and Ant III organ, Chen & Christiansen (1993) [43]; dorsal body Mc, Szeptycki (1979) [47] modified by Jordana & Baquero (2005), Zhang et al. (2011) and Jordana (2012) [48,49,50].

Abbreviations: Abd—abdomen or abdominal segment I–VI, al—anterolateral s-chaeta, as—anterosubmedial s-chaeta, Ant—antennal or antenna/ae, a.s.l.—above sea level, Mc—macrochaeta/ae, mes—mesochaeta, mic—microchaeta, ms—microsensillum/a, PAO—postantennal organ, psp—pseudopore, Th—thorax, or thoracic segments.

Institutions: MNCN—Museo Nacional de Ciencias Naturales (CSIC), Madrid; MZNA—Museum of Zoology at the University of Navarra, Pamplona, Spain; UMACI—Colección de Insectos de la Universidad de Madeira.

## 3. Results

### 3.1. Faunistic

The most diverse communities were detected in the Cavalum caves I and II (nine species in each cave), and III (eight species), while only three species were found in Landeiros.

### 3.2. Taxonomy

Class Collembola Lubbock, 1870 [52]

Order Poduromorpha Börner, 1913 [53] *sensu* d’Haese 2002 [54]

Superfamily Hypogastruroidea Salmon, 1964 [55] *sensu* Deharveng 2004 [56]

Family Hypogastruridae Börner, 1906 [57]

Genus *Ceratophysella* Börner in Brohmer 1932 [58]

#### 3.2.1. *Ceratophysella gibbosa* (Bagnall, 1940) [59]

Studied material: PORTUGAL, Madeira Island, Machico, Cavalum III, E. Nunes leg.

Cavalum III: P4, 17.xii.2001, one specimen on slide.

Habitat and distribution: *C. gibbosa* is an edaphic and holarctic species, and can be found occasionally in caves [33].

Superfamilia Onychiuroidea *sensu* D’Haese 2002–2003 [54,60]

Family Onychiuridae Lubbock in Börner 1913 [53]

Genus *Deuteraphorura* Absolon, 1901 [61]

#### 3.2.2. *Deuteraphorura* cf. *ghidinii* (Denis, 1938) [28]

Studied material: PORTUGAL, Madeira Island, Machico. Cavalum I: P12, 28.viii.2002, one specimen on slide. Cavalum II: C, 24.iv.2002, one specimen on slide; P3, 28.i.2002, one specimen on slide; P3, 28.viii.2002, four specimens on slide; P4, 25.ii.2002, four specimens on slide; P4, 27.iii.2002, one specimen on slide. Cavalum III: C, 25.i.2002, one specimen on slide; C, 28.i.2002, one specimen in ethyl alcohol; P1, 18.ix.2002, two specimens in ethyl alcohol; P3, 17.ix.2001, one specimen on slide; P6, 1.iii.2002, six specimens on slide; P10, 1.iii.2002, two specimens on slide and other two in ethyl alcohol. Landeiros: C, 28.viii.2002, three specimens on slide; C, 15.vii.2002, three specimens on slide. All E. Nunes leg.

Habitat and distribution: *D. ghidinii* (Figure 3) is distributed throughout the Mediterranean region and can be found in both cave and edaphic habitats [33].

##### Remarks

The specimens collected present the same dorsal pseudocelar formula (32/033/33354) and the same ventral pseudocelar formula on Abd I–IV (2212) as *D. ghidinii*, but they differ from this species by the absence of ventral pseudocelli on Th I–III (011 pseudocelli in *D. ghidinii*).

The specimens from Machico caves may represent an undescribed species, but a redescription of the *D. ghidinii* type material would be needed to determine if the morphological differences noted here are fixed.

Genus *Onychiurus* Gervais, 1841 [62]

#### 3.2.3. *Onychiurus* sp.

Studied material: PORTUGAL, Madeira Island, Machico, Landeiros: C, 28.viii.2002, one specimen on slide; C, 15.vii.2002, two specimens on slide. E. Nunes leg.

##### Remarks

The specimens collected appears to represent a closely related species to *O. circulans* as noted by other authors [31,63]. Unfortunately, the state of preservation of the specimens does not allow a precise description.

Order Neelipleona Massoud, 1971 [64]

Family Neelidae Folsom, 1896 [65]

#### 3.2.4. *Neelus serratus* Jordana & Baquero sp. nov.

http://zoobank.org/2588E654-2A14-44AC-9630-8C88240037B5, accessed on 9 May 2023.

##### Type Locality

Cavalum I, II and III, Machico, Madeira Island, Portugal.

##### Type Material

Holotype. Cavalum I: female on slide, sample C, 25.i.2002, E. Nunes leg. Paratypes (all E. Nunes leg.). Cavalum I: same data as holotype, four specimens in ethyl alcohol (+glycerine); P2, 24.iv.2002, one female on slide. P12, 25.ii.2002, one female on slide P13, 25.i.2002, two females on slide. Cavalum II: P3, 19.xi.2001, one female on slide; P3, 28.viii.2002, one female on slide. Cavalum III: C, 25.ii.2002, one female on slide; and Holotype in MZNA; paratypes in MZNA and UMACI.

##### Etymology

The specific name refers to the anterior labral chaetae that are serrated.

##### Diagnosis

Head. Labrum: chaeta a_1_ externally serrate (o micro ciliate), without conspicuous teeth; a_2_ similar to a_1_. Postlabial basal axial chaetae broad, wider than distal ones. Length ratio of Ant III/Ant IV = 1/1.25. Ventral manubrium with 3 + 3 chaetae (one proximal and two distal by side). Legs: claw elongated.

##### Description

Length 0.7–1.0 mm (head included) (n = 8). *Habitus* typical for the genus (Figure 4A). Body color whitish. Cuticle finely granulated. Integumentary channels not observed on head and thorax (probably by the conditions of the specimens).

Head. Length/width: 0.22 mm/0.19 mm (n = 8). Antero dorsal area with 9 + 9 ordinary chaetae. Four prelabral chaetae, and 5,5,4 labral chaetae; anterior chaetae a1 and a2 thick, curved and finely serrate externally; medial chaetae (m-row) thicker than those in p-row, not spine-like; p-row smooth (Figure 4B). Maxillary palp simple, with one enlarged terminal chaeta with a small external process, and a long interior process (probably the reduced lamina), and a basal thick chaeta (Figure 4C). Labium: basomedial field of labium with 4 + 4 chaetae; post-labial chaetae: 3 + 3 (2 + 2 anterior, similar in length: 0.018 mm; 1 + 1 posterior longer (0.022 mm), thickened and straight (Figure 4D). Mandible with five and four asymmetrical apical teeth respectively (Figure 4E). Maxilla as in Figure 4F. Head chaetotaxy in Figure 4G.

Antenna (Figure 5). All antennal segments distinctly separated. Length of antenna, 0.14–0.18 mm (n = 5). Ant I with two small chaetae (three in one specimen); Ant II with one medial chaeta, 4–5 dorsal external apical chaetae (in a whorl), and a ventral apical chaeta; Ant III with 13 normal chaetae (Figure 5, a) in five whorls (distribution from base to apex: 1, 1, 3, 2 and 6), five apical s-chaetae, and an additional thin chaeta (similar to those on Ant IV, named as type c in Figure 2); Ant IV with seven normal chaetae (Figure 5, a), five long with blunt tip (Figure 5, b), 14 thin and small (Figure 5, c), two apical small with blunt tip (Figure 5, d), two sensory chaetae (Figure 5, e): Sy and Sx, and the subapical organite (Or) with variable morphology.

Body. Thorax dorsally covered by microchaetae (0.002 mm). “τ” and “s_3_” chaetae not observed. Posterior dorsal abdominal covered homogeneously with small microchaetae (0.002 mm) (Figure 6A). Neosminthuroid chaeta present on lateral Abd IV sternum. The conservation status of the specimens has not allowed a detailed description of the sensory fields of the thorax, abdomen, and base of the legs. Retinaculum with 3 + 3 apical teeth, without chaetae. Ventral tube with 2 + 2 apical chaetae. Furca well developed; length of manubrium, dens (dp and dd), and mucro: 0.080 mm, 0.029 mm, 0.100 mm, and 0.088 mm, respectively. Manubrium posteriorly with 3 + 3 chaetae, lateral shorter than axial (Figure 6B). Dens dp with 2 + 2 posterior chaetae, lateral shorter than axial; dens with a posterior distal chaeta, E_3_, E_2_ and E1 as spines; externally with AE_1_ big and not articulated (spinous process); internally J_1_ and J_2_ as spines, and AJ_1_ as a spinous process but smaller than the external one (Figure 6C).

Legs. As in Figure 6D–F, with some variations between specimens. Tibiotarsus of leg III with four chaetae with aspect of spines (interior). Claw I longer than II, and II longer than III; tooth on internal lamella on 50–60% of distance of total claw length from basis; external basal lamellae (anterior and posterior) thin, 20–30% of the length of total claw.

##### Habitat

This species was found in the three caves of Cavalum, predominantly on compact rocky substrate associated with water seepage. In Cavalum I it was found in a narrow corridor on a dry sandy substrate where there was a noticeable air current.

##### Remarks

One of the most valuable characteristics to differentiate the species of this genus is the shape of the labrum chaetae a_1_ and a_2_ (row “a”). These characteristics are useful to distinguish between the different species, allowing the direct identification of some species and facilitating the comparison among the remaining (Figure 7).

The new species shares 3 + 3 chaetae in the ventral manubrium with *N. murinus* Folsom 1896 [65], *N. labralisetosus* Massoud & Vannier, 1967 [66], and *N. fimbriatus* Bretfeld & Trinklein, 2000 [67], but it can be differentiated by the following characteristics: from *N. murinus* for the different shape of the labral chaetae a_1_ and a_2_ (row “a”), the thickened axial basal postlabial chaeta, and by some characteristics on the antenna such as the number of normal chaetae and sensilla on Ant IV; from *N. labralisetosus* it is differentiated by the shape of the a_1_ and a_2_ labral chaetae, and by the number of normal chaetae and sensillae on Ant IV. *N. fimbriatus* is more similar to the new species in the labral chaetae of row “a”, although a1 has a basal tooth that the new species does not; in addition, *N. fimbriatus* has a very special morphology for the axial basal postlabial chaeta (feathered-shaped), and Ant IV has a special triangular chaeta that is not present in any other *Neelus* species. Compared with *N. murinus*, the length of the antenna about the head stands out in the new species (ratio 0.7 vs. 0.4 in *N. murinus*); as previously stated, it also has a different appearance in labral chaetae row “a”. It also lacks special chaetae on the abdomen, and it would seem similar about the chaetae (or microchaetae) on the abdomen, although, in the new species, they are restricted to the posterior half and are more numerous.

Ordo Entomobryomorpha Börner, 1913 [53], *sensu* Soto Adames et al., 2008 [68]

Superfamily Entomobryoidea Womersley, 1934 [69]

Family Orchesellidae Börner 1906 [58] *sensu* Zhang et al., 2019 [70]

Subfamily Heteromurinae Absolon & Kseneman, 1942 [71] *sensu* Zhang & Deharveng 2015 [51]

Genus *Heteromurus* Wankel, 1860 [72]

#### 3.2.5. *Heteromurus major* (Moniez, 1889) [73]

Studied material: PORTUGAL, Madeira Island, Machico, Cavalum III: P14, 17.xii.2001, one specimen on slide. E. Nunes leg.

Habitat and distribution

Cosmopolitan species, can be found occasionally in caves [33]

Family Entomobryidae Schäffer, 1896 [74]

Subfamily Entomobryinae Schäffer, 1896 [74] *sensu* Zhang & Deharveng 2015 [51]

Genus *Coecobrya* Yosii, 1956 [75]

#### 3.2.6. *Coecobrya decemsetosa* Jordana & Baquero sp. nov.

http://zoobank.org/3206A092-E377-4C5D-974F-402004787F10, accessed on 9 May 2023.

##### Type Locality

Cavalum I II and III, Machico, Madeira Island, Portugal.

##### Type Material

Holotype: Cavalum II, female on slide, P7, 28.i.2002, E. Nunes leg. Paratypes. Cavalum I: C, 25.ii.2002, two specimen each on slide and 11 in ethyl alcohol; C, 21.vi.2002, one specimen each on slide; P13, 24.iv.2002, one specimen on slide and four in ethyl alcohol; P15, 28.viii.2002, one specimen on slide. Cavalum II: C, 1.viii.2002, one specimen on slide and one in ethyl alcohol; C, 21.xi.2002, one specimen on slide; C, 19.x.2001, one specimen on slide and one in ethyl alcohol; P7, 24.iv.2001, one specimen on slide and one in ethyl alcohol; P7, 28.i.2002, two specimen on slide and seven in ethyl alcohol; P7, 28.viii.2002, one specimen on slide and two in ethyl alcohol; Cavalum III: P1, 24.iv.2002, one specimen on slide. All E. Nunes leg. Holotype in MZNA; paratypes in MNCN, MZNA and UMACI.

##### Etymology

The name refers to the ten dorsal central Mc on Abd IV.

##### Diagnosis

Body length large, up to 2.3 mm; only one post-labial chaetae ‘x’ present; claw with four teeth (two paired and two unpaired); empodium acuminate with a big tooth on outer edge; Abd II with 4 + 4 Mc; Abd IV with 5 + 5 central and 3 + 3 lateral Mc.

##### Description

Body length up to 2.3 mm. Body yellow-white.

Antenna 1.74 times as long as cephalic diagonal. Antennal segments ratio I:II:III:IV = 1:1.7:1.4:3. Ant II and IV with normal ciliated chaetae and three types of sensilla: short, curved and straight (Figure 8A); Ant IV not annulated under microscope, with subapical organite as Figure 8B; Ant III organ with five sensilla: two rod-like (some expanded) and three small sensilla (Figure 8C).

Without eyes. Clypeal chaetae (Zhang et al., 2016) as in Figure 8D. All labral chaetae smooth (Figure 8E). Papilla E with four chaetae and lateral process surpassing de papilla (Figure 8F); labial posterior row chaetae all smooth (-mrell), r smaller (half m). Cephalic groove with eight chaetae, only the fifth and the eighth ciliate; numerous chaetae smooth and ciliated on the ventral part of the head and with a small ‘x’ chaeta (Figure 8G). Dorsal cephalic chaetotaxy with three An, A_0_, A_2_, A_3_ and A_5_ as Mc; M_1_–M_2_, M_4_, sutural row: S_0_–S_5_ as Mc; S_5i_ and S_4i_ as mes, Ps_2_ present as Mc; three Mc in Gr. II (Figure 8H).

V-shaped trochanteral organ with 13 straight smooth spiny chaetae (Figure 8I). All tibiotarsal chaetae ciliate with the exception of two rows of smooth chaetae. Claw with four teeth, first pair at 50% from basis, first unpaired at 70%, and a small subapical one; dorsolateral wing teeth present at a level of internal paired teeth, dorsal tooth absent. Empodium outer edge smooth, with a big external tooth. Tenent hairs spatulated, the same length as the claw (Figure 8J).

Abd IV 3.1 times as Abd III in length along dorsal midline. Ventral tube anteriorly with about five ciliate chaetae on each side; each lateral flap with 10 smooth chaetae (Figure 8K). Manubrium with many smooth straight chaetae (Figure 8L). Manubrial plate with two pseudopores and two ciliate chaetae. Distal smooth part of dens as mucro in length. Mucro with one tooth, and basal spine reaching to the tip of mucro (Figure 8M).

Body macrochaetotaxy (Figure 9A,B). Th II with three medio-medial (m_1_, m_2_, m_2i_), three mediolateral (a_5_, m_4_, m_4p_), 12 posterior on the p_1_-p_3_ group, and 11 on the p_4_ group Mc; anterior lateral ms and as, and posterior sp sensillum present. Th III with about 25 Mc (some could be mes). Abd I with eight Mc; ms antero-external and an S-chaeta present. Abd II without anterior Mc, three ‘m’ central Mc (m_3_, m_3e_, and m_3ea_), and one (m_5_) lateral Mac. Abd III with one central Mc (m_3_) and three (a_7_, am_6_, and p_6_) lateral Mc; two S-chaetae presents (as and al). Abd IV with five central Mc (A_4_, A_6_, B_4_-B_6_), three lateral Mc (E_2_, E_3_, and E_4_), and at least six long and two short S-chaetae. Abd V with three S-chaetae. Macrochaetotaxy simplified formula following Jordana & Baquero 2005, and Jordana 2012: 310221b/33/03/001/00212C/00111L.

##### Habitat

Apparently, it is a cave-restricted species because it has been captured in the tree Cavalum caves of the Madeira Island.

##### Remarks

Only three species of *Coecobrya* have four teeth on the claw: *C. montana* (Imms, 1912) [76] *sensu* Zhang, Deharveng & Chen, 2009 [77], and *C. submontana* Stach, 1960 [78]. This new species, *C. montana* (from India), has a basal dorsal tooth on the claw, and this is the only known difference with the new species since its chaetotaxy is unknown. *C. submontana* (Afganistan), which has been studied after requesting some specimens deposited in the Stach`s collection of PAN (Polska Akademia Nauk, Warszawa, Poland), from a cave near Kabul, has a short basal spine on the mucro, and the dorsal tooth of the claw in an intermediate position between the base and the paired teeth of the inner edge. With 10 central Mc on Abd IV, there are eight species in the world: *C. ishikawai* Yosii, 1956 [75], *C. huangi* Chen & Ma, 1998 [79], *C. khaopaela* Zhang & Jantarit, 2018 in Zhang et al., 2018 [80], *C. phanthuratensis* Zhang & Jantarit, 2018 in Zhang et al., 2018 [80], *C. qinae* Xu & Zhang, 2015 [81], *C. ranongica* Nilsay & Zhang, 2018 in Zhang et al., 2018 [80], *C. sanmingensis* Xu & Zhang, 2015 [81], and *C. xui* Zhang & Dong, 2014 [82], but all of them have three teeth on the claw.

#### 3.2.7. *Coecobrya octoseta* Jordana & Baquero sp. nov.

http://zoobank.org/774467AD-780F-407E-AA4D-40DBDCC48328, accessed on 9 May 2023.

##### Type Locality

Cavalum I, II, and III caves, Machico, Madeira Island, Portugal.

##### Type Material

Holotype: Cavalum I. Female on slide, P2, 25.ii.2002, E. Nunes leg. Paratypes. Cavalum I: C, 24.iv.2002, one specimen on slide and four in ethyl alcohol; C, 1.viii.2002, one specimen on slide; P1, 25.ii.2002, one specimen on slide and 35 in ethyl alcohol; P1, 24.iv.2002, 1 specimen on slide and 52 specimens in ethyl alcohol (damaged); P1, 28.viii.2002, one specimen on slide; P1, 17.xii.2001, one specimen on slide; P2, 25.ii.2002, one specimen on slide and 12 in ethyl alcohol; P2, 27.iii.2002, one specimen on slide and two in ethyl alcohol; P2, 24.iv.2002, one specimen on slide and one in ethyl alcohol; P3, 28.viii.2002, two specimens on slide; P5, 27.iii.2002, one specimen on slide; P5, 24.iv.2002, one specimen on slide and six in ethyl alcohol. P11, 24.iv.2002, one specimen on slide; P11, 28.viii.2002, one specimen on slide; P14, 28.viii.2002, one specimen on slide; Cavalum II: P1, 25.ii.2002, one specimen on slide; P4, 24.iv.2002, one specimen on slide; P5, 25.ii.2002, one specimen on slide and one in ethyl alcohol; Cavalum III: C, 25.ii.2002, one specimen on slide and one in ethyl alcohol; C, 18.ix.2002, one specimen on slide; P1, 26.i.2002, one specimen on slide; P2, 26.i.2002, two specimens on slide and eight in ethyl alcohol; All E. Nunes leg. Holotype in MZNA; paratypes in MNCN, MZNA and UMACI.

##### Etymology

The name refers to the eight dorsal central Mc on Abd IV.

##### Diagnosis

Size up to 1.6 mm (Figure 10); two ventral labial chaetae x present; claw with three teeth (two paired and one unpaired); empodium truncate with a big tooth on outer edge; Abd II with 4 + 4 central Mc; Abd IV with 4 + 4 central and 5 + 5 lateral Mc.

##### Description

Body length up to 1.6 mm (n = 3). Body yellow-white.

Antenna 1.83 times (n = 3) as long as cephalic diagonal. Antennal segments ratio as I:II:III:IV = 1:1.8:1.7:3.1; Ant III organ with five sensilla: two expanded and three small sensilla (Figure 11A); Ant II and IV with normal ciliated chaetae (Figure 11(B_1_)) and three types of sensilla: short, curved and straight (Figure 11(B_2_–B_5_)); Ant IV not annulated under microscope, with a special chaeta (Figure 11C), subapical organite (Figure 11D), and a straight terminal chaeta (Figure 11E).

Without eyes. Clypeal chaetae as in Figure 9F. Labral chaetae all smooth (Figure 11G). Papilla E with four guard chaetae and lateral process surpassing de papilla (Figure 11H); labial posterior row chaetae all smooth (mrell), r smaller (only slightly shorter than m). Cephalic groove with six chaetae (fourth and fifth ciliate); numerous chaetae smooth and ciliated on the ventral part of the head and with two small x chaeta (Figure 11I). Dorsal cephalic chaetotaxy with three An, A_0_, A_2_, A_3_ and A_5_ as Mc; M_1_ M_2_, M_4_, sutural: S_0_, S_1_, S_2_, S_3_, S_4_ and S_5_ as Mc; S_5i_ and S_4i_ as mes, Ps_2_ present as Mc; two Mc in Gr. II (Figure 11J).

Trochanteral organ with 14 straight smooth spiny chaetae (Figure 11K). Tibiotarsal chaetae ciliate and two rows of smooth chaetae. Claw with three teeth, first pair 30% from basis, unpaired at 50%; dorsolateral wing teeth present, before paired teeth, dorsal tooth basal. Empodium truncate in shape, outer edge smooth, and with a big external tooth. Tenent hairs acuminate, same length as claw; smooth terminal chaeta very thin (Figure 11M).

Abd IV 3.5 times as Abd III in length along dorsal midline. Ventral tube anteriorly with about three ciliate chaetae on each side; each lateral flap with seven smooth chaetae. Manubrium 0.8 times shorter than dens, with many smooth straight chaetae. Manubrial plate with two pseudopores and two ciliate chaetae. Distal smooth part of dens only a little longer as mucro in length. Mucro falcate, and basal spine normal, reaching to the tip of mucro (Figure 11N).

Body macrochaetotaxy (Figure 12A,B). Th II with four medio-medial (m_1_, m_1p_, m_2_ and m_2i_) (T1 or group I), three medio-lateral (a_5_, m_4_ and m_4p_) (T_2_ or group II), 10 posterior chaetae on groups III–IV (six Mc and four mes), 10 on group V and two on group VI; anterior lateral ms and as, and posterior sp sensilla present. Th III with about 25 chaetae (10–14 as Mac). Abd I with nine Mc; ms antero-external and an s-chaeta present. Abd II without anterior Mc and three ‘m’ central Mc (m_3_, m_3e_, and m_3ea_), and one (m_5_) lateral Mac. Abd III with only one central Mc (m_3_), and three (a_7_, am_6_ and p_6_) lateral Mc; two S-chaetae presents (as and al). Abd IV with four central Mc (A_3_–A_6_, B_3_), five lateral Mc (E_2_, E_2p_, F_2_, F_3a_ and another latero-posterior one), and approximately 10 S-chaetae, one shorter (on the typical position). Abd V with three S-chaetae. Macrochaetotaxy simplified formula following Jordana & Baquero 2005 [48] and Jordana 2012 [50]: 310221b/34/03/001/00211C/00212L.

##### Habitat

This species is common in all three Cavalum caves and is associated with places with water infiltration and puddles. However, in Cavalum II, it was found close to debris.

##### Remarks

Four species of this genus have the following combination of characters: 0 eyes, eight central Mc in Abd IV, 4(5) + 4(5) lateral Mc, and pointed tenent hair: *Coecobrya akiyoshiana* Yosii 1956 [75], *C. nupa* (Christiansen & Bellinger 1992) [83], *C. oligoseta* (Chen & Christiansen 1997) [44], and *C. tenebricosa* (Folsom 1902) [84]. This species, in many characteristics, resembles *C. aitorerere* Bernard, Soto-Adames & Wynne 2015 [85], but it is separated by the thickness of the Mc, which is evident in the drawings, and for having three lateral Mc following the pattern VII [44]. *C. nupa* is well differentiated by having two and five Mc in the T1 and T2 areas on the Th II; it also has very basal internal teeth of the claw, although less than the following species. *C. akiyoshiana* has two Mc in areas T1 and T2 in Th II, while the new species has 3(4) and three Mc in those areas. In addition, the paired and unpaired teeth in the inner edge of the claw are very basal (14 and 20% about the claw base) while they are almost medial in the new species. *C. oligoseta* has four Mc in the T2 area of Th II but three Mc in the new species and five lateral Mc as pattern V [44], while the new species has them in pattern IV. However, the two most similar species are *C. tenebricosa* and *C. aitorerere*.

Some differences are observed between these three species: *C. tenebricosa* has 9–13 Mc in group I of the metanotum, *C. aitorerere* has eight, and the new species described have six (two of them smaller). Group II: *C. tenebricosa* 9–10, *C. aitorerere* eight, and the new species 12. Th II Groups III–VI: *C. tenebricosa* 6-2-9-2, and new species: 4-5-10-2. Th III Groups I–II: *C. tenebricosa* 10–9, and the new species: 5–10. There are broad Mc in tibiotarsi in *C. aitorerere*. Claw, lateral teeth: in *C. tenebricosa* is basal and in the new species is at the paired teeth level. Abd III, lateral Mc number: *C. tenebricosa* two, *C. aitorerere* two, and new species three. Abd IV, lateral Mc number: *C. tenebricosa* 4–5, *C. aitorerere* three, and the new species five. Long sensilla above bothriotrichum: *C. tenebricosa* two, and the new species four. We believe that there are enough differences to separate this species as new.

Genus *Sinella* Brook, 1882 [86]

#### 3.2.8. *Sinella duodecimoculata* Jordana & Baquero sp. nov.

http://zoobank.org/C2E2F1EA-C422-4667-8BB0-E92DFC802ADF, accessed on 9 May 2023.

##### Type Locality

Cavalum I cave, Machico, Madeira Island, Portugal.

##### Type Material

Holotype: female on slide, P11, 27.iii.2002, E. Nunes leg. Paratypes. Cavalum I: P14, 13.xi.2001, seven specimens on slide; P14, 19.xi.2001, two specimens in ethyl alcohol; P14, 17.xii.2001, one specimen on slide; P4, 27.iii.2002, three specimens on slide. P11, 27.iii.2002, one specimen on slide (the same of holotype); P13, 27.iii.2002, one specimen on slide; P13, 28.viii.2002, one specimen on slide. All E. Nunes leg. Holotype in MZNA; paratypes in MNCN, MZNA and UMACI.

##### Etymology

Named after the number of eyes.

##### Diagnosis

This new species has 6 + 6 eyes and claws with three teeth on the inner edge. There are seven Mc in Abd II between bothriotricha; six central Mc on Abd IV, and the absence of a terminal vesicle in Ant IV.

##### Description

Body length up to 2.7 mm. Body yellow-white.

Antenna 2.3 times as long as cephalic diagonal. Antennal segments ratio as I:II:III:IV = 1:2.2:2.2:3. Long striated straight chaetae on Ant I–IV, Ant II, and IV with normal ciliated chaetae and sensilla of three types: short, curved, and straight; Ant IV with subapical organite as Figure 13A; Ant III organ with five sensilla: two rod-like and three small sensilla (Figure 13B).

Eyes 6 + 6; ABCDEF, without G and H. Papilla E with three chaetae, lateral process surpassing de papilla (Figure 13C); labial chaetae smooth (mmrell), m_1_ and r similar in size and half of m_2_ (Figure 13D). Cephalic groove with nine ciliated chaetae and with numerous chaetae on the ventral part of the head.

Trochanteral organ with 20–22 straight smooth spiny chaetae distributed in two arms of seven and ten chaetae respectively, and five–six chaetae between the arms (Figure 13F). All tibiotarsal chaetae ciliate with striations. Empodium outer edge smooth. Tenent hairs spatulated, longer than empodium (Figure 13G). Claws with three teeth, first pair at 50% from basis, unpaired one at 70%. Dorsal teeth basal.

Abd IV 3.5 times as Abd III in length along dorsal midline. Manubrium without smooth straight chaetae. Manubrial plate with two pseudopores and seven ciliate chaetae. Distal smooth part of dens 3.0 times of mucro in length. Mucro bidentate with two teeth, anteapical bigger than apical one; basal spine surpassing the tip of subapical tooth a little (Figure 13H).

Body macrochaetotaxy (Figure 13E and Figure 14A,B). Dorsal cephalic macrochaetotaxy with five An, A_0_, A_2_, A_3_ and A_5_, M_1_–M_4_, sutural row (S_0_–S_5_) and Ps_2_ as Mc. Th II with four medio-medial (m_1_, m_2_, m_2i_ and m_2i2_), five medio-lateral (a_5_, m_4_, m_4i_, m_4p_ and m_5_), and 20 posterior Mc, with an anterior lateral ms and s sensillum (posterior sensillum not seeing). Th III with 30 Mc; m_5i_ and a_6i_ as Mc. Abd I with 10 Mc; ms antero-external and an S-chaeta. Abd II with two anterior Mc (a_2_–a_3_) and five ‘m’ Mc (m_3_, m_3ep_, m_3e_, m_3ei_ and m_3ea_) central and one (m_5_) lateral Mc. Abd III with two Mc (a_2_ and a_3_) anterior to the bothriotrichum, m_3_ as Mc, and three (am_6_, pm_6_, p_6_) lateral Mc; one ms and two S-chaetae present. Abd IV with six central Mc (A_3_, A_5_–A_6_, B_4_–B_6_), three lateral Mc (E_2_, E_3_ and D_3_), and 11 S-chaetae; most S-chaetae much longer than those on anterior terga. Abd V with three S-chaetae.

Macrochatotaxy simplified formula [48,50]: 410311b/45/25/021/01122C/00111L.

##### Habitat

Found on a cave in Madeira island. This species was found in a single chamber of Cavalum I where there are draughts, but no infiltrations occur.

##### Remarks

Only five *Sinella* species have six eyes: *Sinella jaldaparaensis* Mandal, Suman & Bhattacharya, 2019 [87], *S. jugoslavica* Loksa & Bogojevic, 1970 [88], *S. pulcherrima* Agrell, 1939 [89], *S. siva* (Imms, 1912) [76], and the new species described here. *S. siva* has four teeth on the claw, *S. pulcherrima* has two while the new species has three, both have paired teeth on the inner edge of the claw very basal (10% in relation to the claw base), while *that S. jugoslavica* and the new species have them in 50% of the inner edge of the nail.

This new species is close to *S. jugoslavica*, sharing with it 6 + 6 eyes and a claw with three teeth on the inner edge (with similar distribution along the inner edge of the claw), but differs in six Mc between bothriotricha in Abd II (half segment) while there are seven in the new species; 18 central Mc on Abd IV (half segment) whereas there are only six in the new species, and the presence of a terminal vesicle in Ant IV, absent in the new.

The only species of the genus reported from Madeira is *S. pulcherrima*, which has the following characters different from the new species: the only teeth present on the inner edge (paired) are very basal; although it also has six eyes, they are in a different position (ABCDEH); and the distal tooth of the mucro is larger than the basal one. Very few characteristics have been described in *S. jaldaparaensis* and *S. siva*, but there are enough for differentiation: *S. siva* has two teeth on the internal edge of the claw at 25% odd the base, while the new species has the paired teeth at 54% from the base, and *S. jaldaparaensis* has four teeth, which is quite different to the new species.

Genus *Lepidocyrtus* Bourlet, 1839 [90]

#### 3.2.9. *Lepidocyrtus curvicollis* Bourlet, 1839 [90]

Studied material: PORTUGAL, Madeira Island, Machico, Cavalum II: P3, 18.xi.2001, one specimen in ethyl alcohol; P9, 19.xi.2001, one specimen on slide; P4, 25.ii.2002, one specimen on slide. Cavalum III: C, 17.xii.2001, one specimen on slide. All E. Nunes leg.

##### Habitat and Distribution

*L. curvicollis* has a holarctic distribution and can be found in both edaphic and cave habitats [91,92]. On Madeira Island this species has already been cited in the cavities Cavalum I, II, and III [33], although some of these previously cited specimens most likely belong to the species *L. flexicollis*.

#### 3.2.10. *Lepidocyrtus flexicollis* Gisin, 1965 [93]

Studied material: PORTUGAL, Madeira Island, Machico, Cavalum I: P11, 17.xii.2001, one specimen on slide. Cavalum II: P3, 19.xi.2001, one specimen on slide; P9, 19.xi.2001, one specimen on slide; P10, 19.xi.2001, one specimen on slide; P4, 25.ii.2002, one specimen on slide; P11, 28.viii.2002, one specimen on slide. Cavalum III: C, 17.xii.2001, one specimen on slide; P7, 15.xi.2001, eight specimens on slide and four in ethyl alcohol; P14, 17.xii.2001, 12 specimens on slide and nine in ethyl alcohol (Figure 15). All E. Nunes leg.

##### Habitat and Distribution

*L. flexicollis* is distributed throughout the southeast of the Iberian Peninsula and the Canary Islands and can be found in edaphic habitats and caves [94]. Cited by Arbea et al. (2021) [33] as *L. curvicollis*. This publication represents the first record of the species for Madeira.

Genus *Entomobrya* Rondani, 1861 [95]

#### 3.2.11. *Entomobrya pazaristei* Denis, 1933 [96]

Studied material: PORTUGAL, Madeira Island, Machico, Cavalum II: P6, 17.xii.2001, one specimen on slide. Cavalum III: C, 17.xii.2001, one specimen on slide and other in ethyl alcohol; P7, 15.xi.2001, three specimens on slide; P14, 17.xii.2001, 10 specimens on slide and nine in ethyl alcohol. All E. Nunes leg.

##### Habitat and Distribution

The species (Figure 16) has been reported from caves in Croatia, Romania, Spain, Portugal, Balearic Is., and Madeira [33].

Order Symphypleona Börner, 1901 [97]

Superfamilia Katiannoidea Bretfeld, 1994 [98]

Family Arrhopalitidae Stach, 1956 [99] *sensu* Bretfeld, 1999 [100]

Genus *Pygmarrhopalites* Vargovitsh, 2009 [101]

#### 3.2.12. *Pygmarrhopalites elegans* (Cassagnau & Delamare-Deboutteville, 1953) [27]

Studied material: PORTUGAL, Madeira Island, Machico, Cavalum I: P3, 24.iv.2002, two specimens on two slides. E. Nunes leg.

##### Habitat and Distribution

Troglophilous species are present in both edaphic and cave habitats of the Iberian Peninsula, Madeira, France, and Croatia [33].

#### 3.2.13. *Pygmarrhopalites* sp. 1

Studied material: PORTUGAL, Madeira Island, Machico, Cavalum I: P3, 28.viii.2002, one specimen on slide. Cavalum II: P1, 28.i.2002, one specimen on slide and four in ethyl alcohol. All E. Nunes leg.

##### Remarks

The state of conservation of the specimens does not allow identification, although it can be said that they do not belong to the species *P. elegans* because they have a completely different female subanal appendage.

Genus *Disparrhopalites* Stach, 1956 [99]

#### 3.2.14. *Disparrhopalites patrizii* (Cassagnau & Delamare Deboutteville, 1953) [27]

Studied material: PORTUGAL, Madeira Island, Machico, Cavalum I: P11, 17.xi.2001, one specimen in ethyl alcohol; PIII, 17.xii.2001, 15 specimens in ethyl alcohol. Cavalum II: C, 1.viii.2002, three specimens on slide. 19.xi.2001: P6, three specimens in ethyl alcohol; P7, one specimen in ethyl alcohol; P8, four specimens in ethyl alcohol; P10, two specimens in ethyl alcohol; P11, one specimen in ethyl alcohol. 15.xii.2001: P17, three specimens in ethyl alcohol. 17.xii.2001: P6, six specimens in ethyl alcohol; P7, one specimen in ethyl alcohol; P8, one specimen in ethyl alcohol; P8, three specimens in ethyl alcohol; P9, four specimens in ethyl alcohol; P10, three specimens in ethyl alcohol; P11, two specimens on slide and one in ethyl alcohol. 28.i.2002: P6, 44 specimens in ethyl alcohol; P7, 30 specimens in ethyl alcohol; P8, 35 specimens in ethyl alcohol; P9, six specimens in ethyl alcohol; P10, five specimens in ethyl alcohol; P11, two specimens in ethyl alcohol. 25.ii.2002: P6, one specimen on slide and 70 specimens in ethyl alcohol; P7, 124 specimens in ethyl alcohol. 27.iii.2002: P6, 10 specimens in ethyl alcohol; P7, 39 specimens in ethyl alcohol; P8, one specimen on slide and 79 specimens in ethyl alcohol; P9, one specimen in ethyl alcohol. 24.iv.2002: P6, 46 specimens in ethyl alcohol; P7, 144 specimens in ethyl alcohol; P9, two specimens in ethyl alcohol; P10, one specimen in ethyl alcohol. 28.viii.2002: P7, one specimen on slide and 39 specimens in ethyl alcohol; P8, 65 specimens in ethyl alcohol; P9, seven specimens in ethyl alcohol; P10, one specimen in ethyl alcohol; P11, two specimens on slide. Landeiros: 18.ix.2002: P4, one specimen in ethyl alcohol (infested with fungi). All E. Nunes leg.

##### Habitat and Distribution

*D. patrizii* is distributed throughout the south of Europe. It is a higrophilic and troglophilic species (Arbea et al., 2021). It has already been cited by Delamare-Debouteville & Bassot (1957) [30] as present in these caves.

## 4. Discussion

Three species of springtails had already been reported in the Cavalum cave complex in the 1950s: *D. patrizii* [27], *D. ghidinii*, and *O. circulans* [31]. Three species of springtail recorded in the present study can be classified as epigeal or edaphic, common in surface habitats and occasionally colonizing cave entrances: *C. gibbosa* (Cavalum III), *E. pazaristei* (Cavalum II, III and IV), and *H. major* (Cavalum III). These caves are mainly inhabited by seven troglophile species: *D.* cf. *ghidinii* (present in the Cavalum and Landeiros caves), *Onychiurus* sp. (exclusive to Landeiros), *S. duodecimoculata* sp. nov. (exclusive to Cavalum I), *L. curvicollis* (Cavalum II and III), *L. flexicollis* (Cavalum I, II and III), *P. elegans* (Cavalum I), and *D. patrizii* (Cavalum I and II, and Landeiros). The four troglobitic species are limited to the caves of Cavalum: (*C. octoseta* sp. nov.) (Cavalum I, II and III), *C. decemsetosa* sp. nov. (Cavalum I, II and III), *Neelus serratus* sp. nov. (Cavalum I, II and III), and *Pygmarrhopalites* sp. 1 (Cavalum I and II).

Machico is a small town interested in valorizing its natural heritage. The exploitation of the caves has always been present as a hotspot of its development but so far, no intervention has ever been made. Fortunately, the size of these caves is not suitable for tourist exploration. Moreover, with the construction of tunnels nearby, it is possible that internal landslides may have occurred and safety issues for visitors may have to be addressed. In order to maintain their fauna in the long term, it would be necessary for the land to be owned and managed by public bodies and access to them to be restricted to a minimum. One way to reconcile tourist development without endangering the natural heritage would be to build a replica museum of a cave elsewhere where visitors could see the geological formations and specimens of its fauna. In order to decide on any possible management plan, it is urgent and necessary to evaluate the current state of the fauna present in all the Cavalum caves and to explore one that has not been studied in this work (Cavalum IV).

## Figures and Tables

**Figure 1 insects-14-00525-f001:**
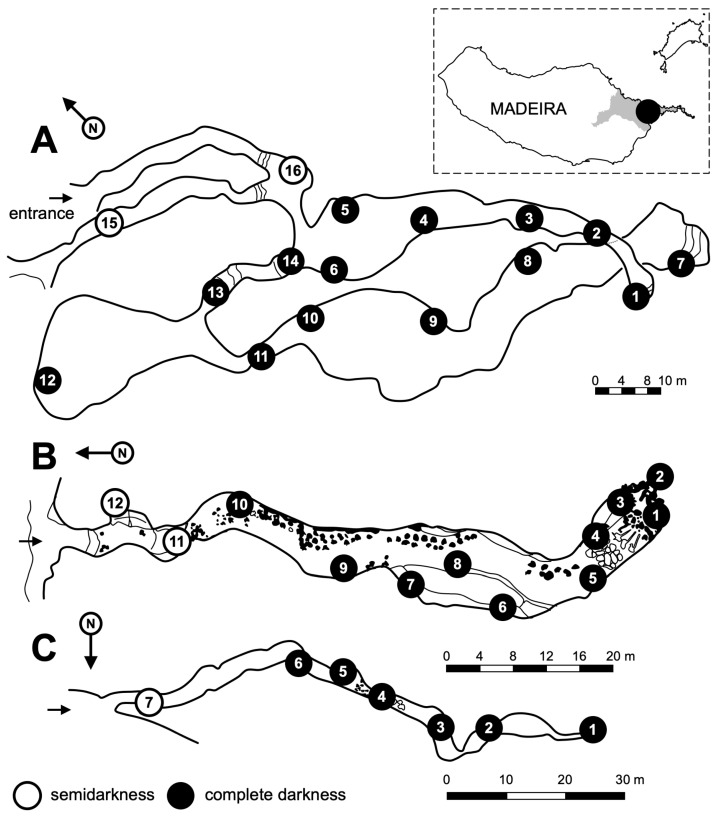
Topographic map of the Cavalum caves studied (adapted from Calandri 1991) [34]. Name of caves and corresponding authors of topographic maps: (**A**) Cavalum I; (**B**) Cavalum II; (**C**) Cavalum III.

**Figure 2 insects-14-00525-f002:**
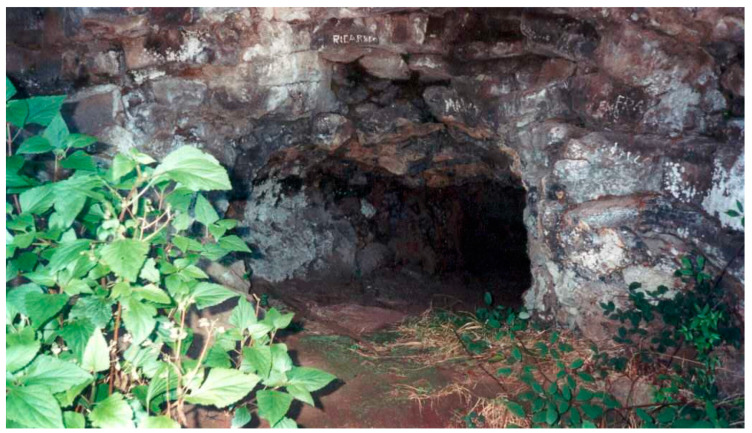
Entrance of Cavalum III (Photo. Elvio Nunes).

**Figure 3 insects-14-00525-f003:**
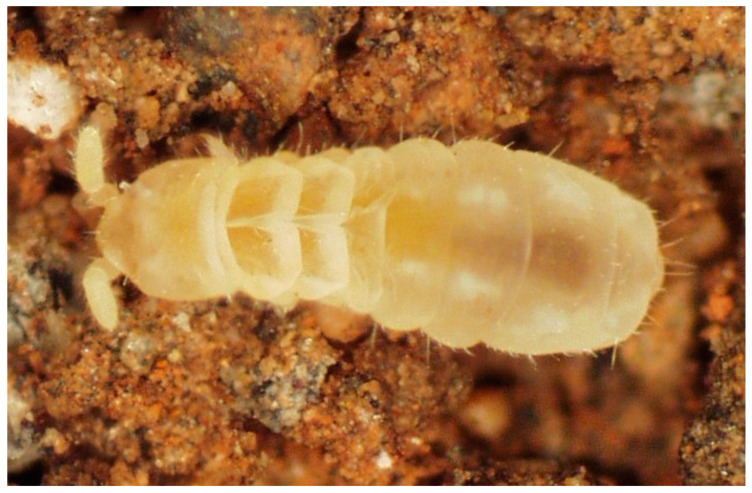
Alive specimen of *Deuteraphorura* cf. *ghidinii* photographed in the lab (photo: Nélio Freitas).

**Figure 4 insects-14-00525-f004:**
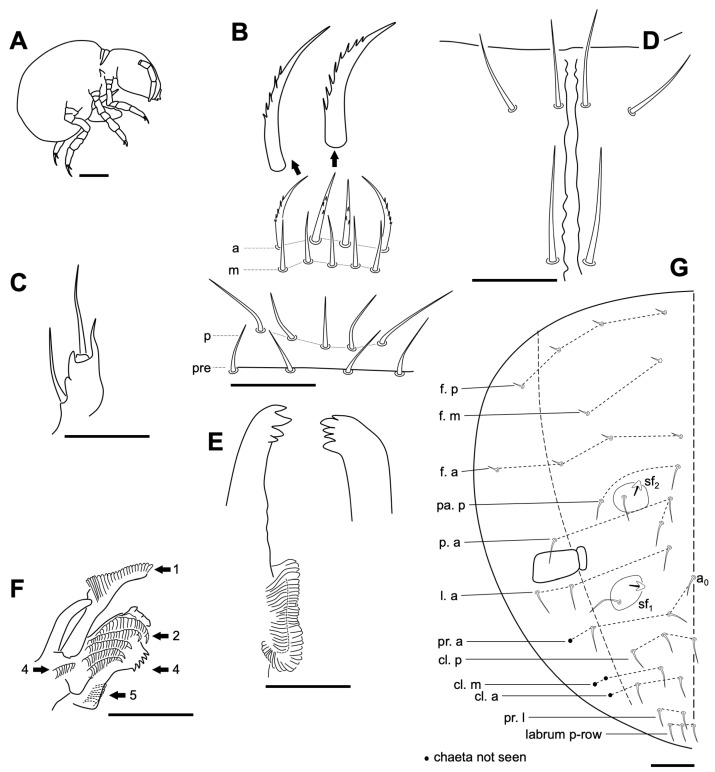
*Neelus serratus* Baquero & Jordana sp. nov. (**A**) habitus of the new species; (**B**) labral chaetae; (**C**) labial papilla E; (**D**) ventral head, distal part of the canal; (**E**) mandibles; (**F**) maxilla (lamellae signed from 1 to 5); (**G**) dorsal head chaetotaxy. Scale bars: 0.02 mm, except for habitus (0.2 mm).

**Figure 5 insects-14-00525-f005:**
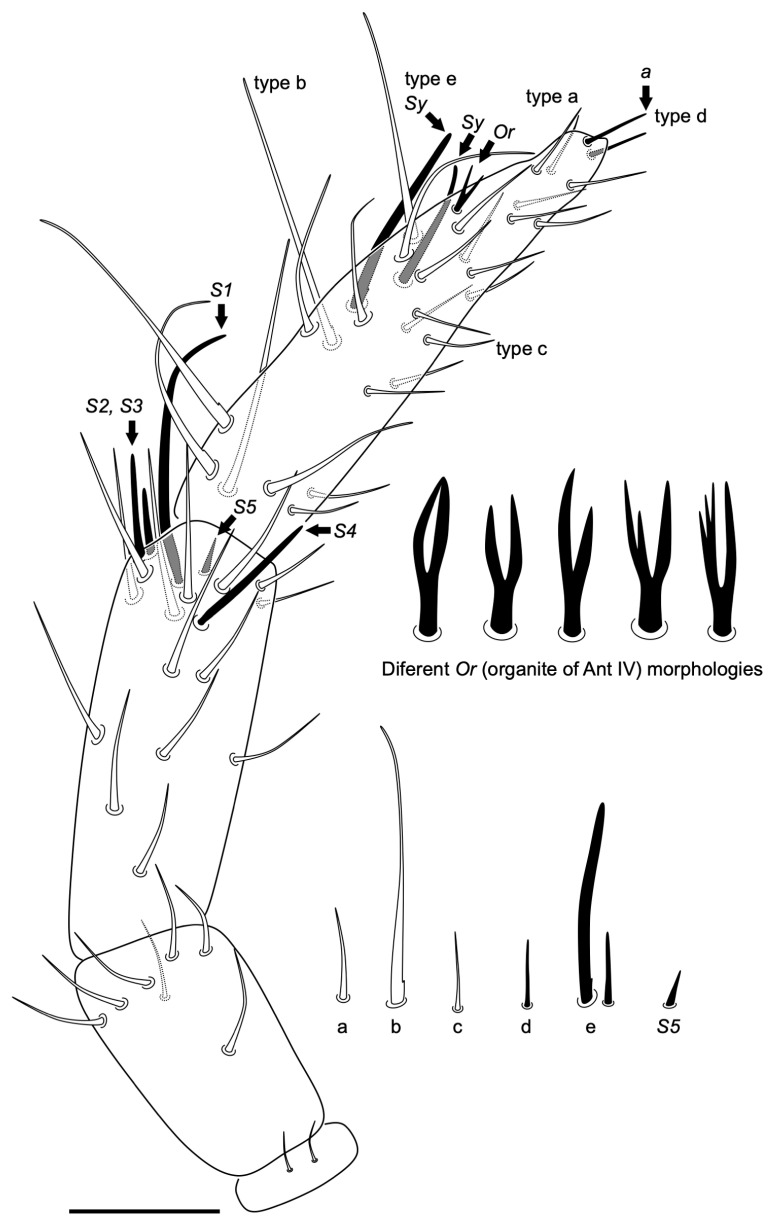
*Neelus serratus* Baquero & Jordana sp. nov. antenna, with detail of the ‘or’ organite and the different shape of some chaetae: a, small normal chaeta; b, big chaeta; c, small thin chaeta (distal area of Ant IV); d, small sensillum; e, other sensillae; S5, small sensillum S5. Scale bar: 0.02 mm.

**Figure 6 insects-14-00525-f006:**
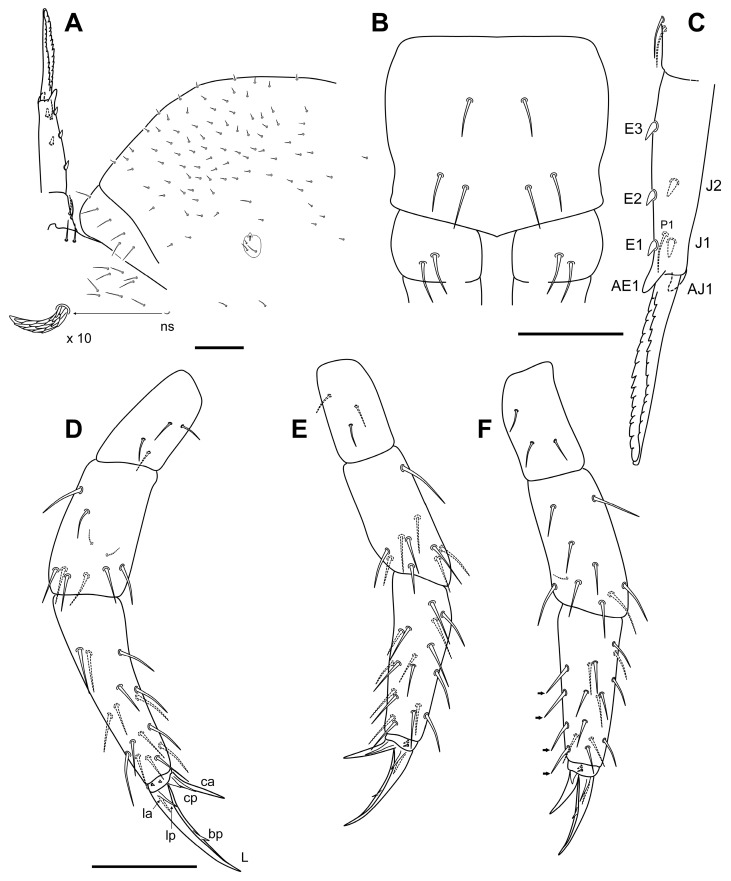
*Neelus serratus* Baquero & Jordana sp. nov. (**A**) dorso-posterior area of body, with part of the furca and detail of the neosminthuroid chaeta (ns); (**B**) *manubrium* and proximal area of *dens*; (**C**) lateral view of the furca; (**D**–**F**) legs 1–3 respectively. Scale bar: 0.05 mm.

**Figure 7 insects-14-00525-f007:**
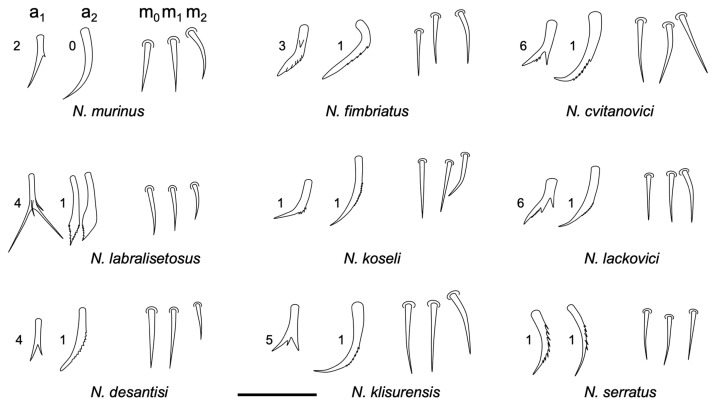
Labral shape of chaetae a_1_ and a_2_ (row ‘a’) of the whole described species of *Neelus*: 0, smooth; 1, external serrated; 2, basal tooth; 3, serrated and with basal tooth; 4, bifurcated; 5, with more than two branches; 6, basal or medial tooth and internally serrated. Scale bar: 0.02 mm.

**Figure 8 insects-14-00525-f008:**
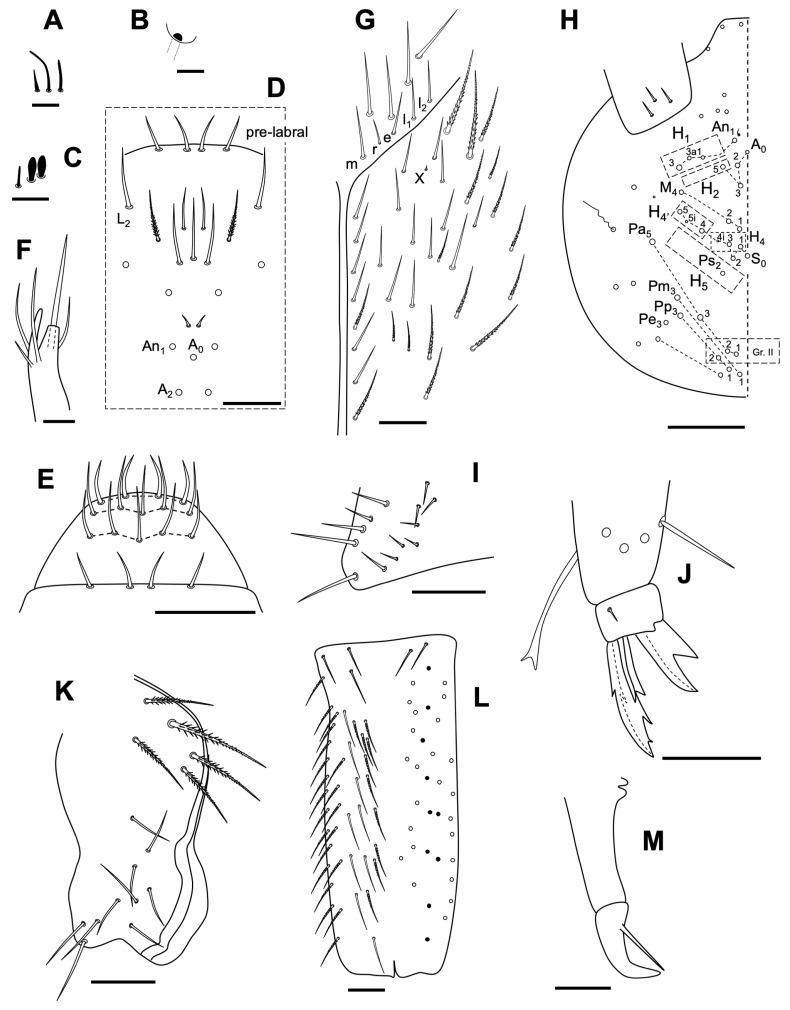
*Coecobrya decemsetosa* Jordana & Baquero sp. nov. (**A**) three different types of sensillae; (**B**) special organite of Ant IV; (**C**) special sensillae of Ant III sensory organ; (**D**) head clypeal area; (**E**) pre-labral and labral chaetae; (**F**) labial papilla E; (**G**) ventral head, labial posterior area; (**H**) dorsal head chaetotaxy; (**I**) trochanteral organ; (**J**) distal area of leg 3; (**K**) ventral tube; (**L**) *manubrium*; (**M**) distal area of *dens* and mucro. The white circles represent ciliated chaetae, and their size is proportional to the size of the alveolus; in Figure 8L the black circles represent smooth chaetae. Scale bars: (**A**–**C**) 0.001 mm; (**D**,**I**–**K**) 0.02 mm; (**E**–**G**,**L**,**M**) 0.025 mm; (**H**) 0.1 mm.

**Figure 9 insects-14-00525-f009:**
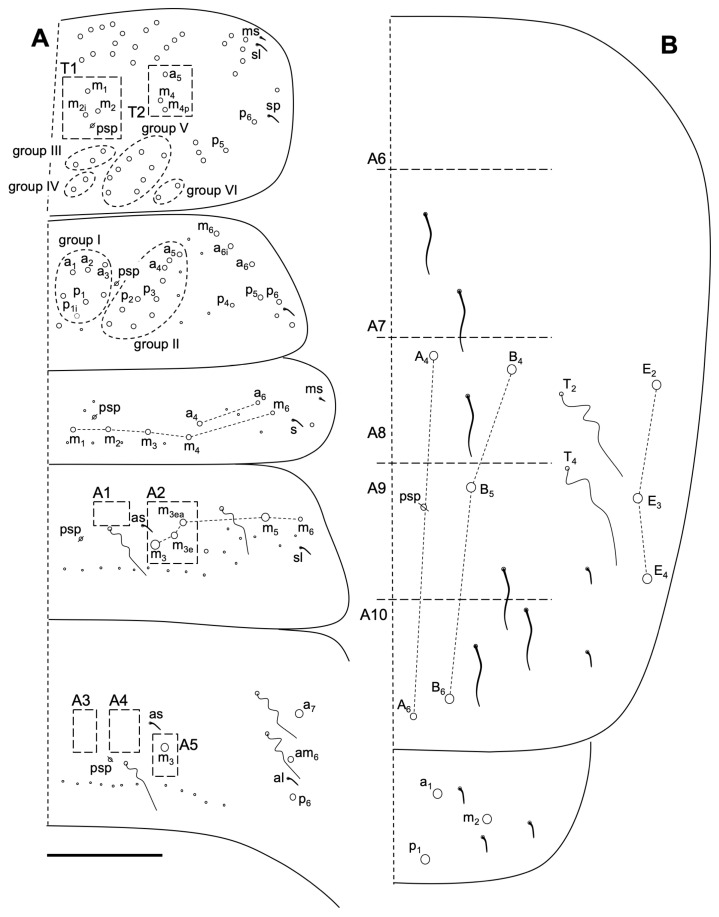
*Coecobrya decemsetosa* Jordana & Baquero sp. nov. dorsal macrochaetotaxy. (**A**) Th II to Abd III; (**B**) Abd IV–V. The white circles represent ciliated chaetae, and their size is proportional to the size of the alveolus. Scale bar: 0.05 mm.

**Figure 10 insects-14-00525-f010:**
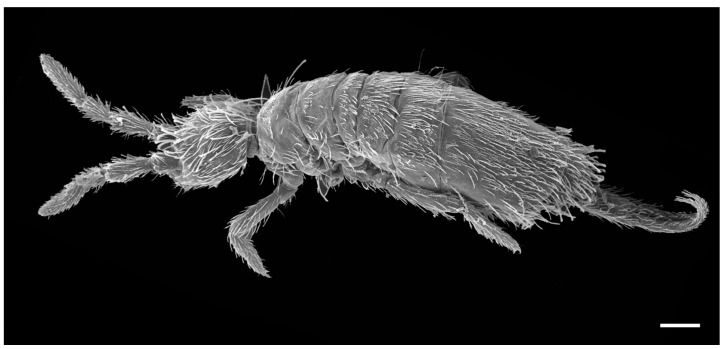
*Coecobrya octoseta* Jordana & Baquero sp. nov., SEM microphotograph of a specimen from Cavalum I (capture data: EHT, 10 kV; WD, 39 mm; detector, SE2. Bar, 0.1 mm. Photo: E. Baquero).

**Figure 11 insects-14-00525-f011:**
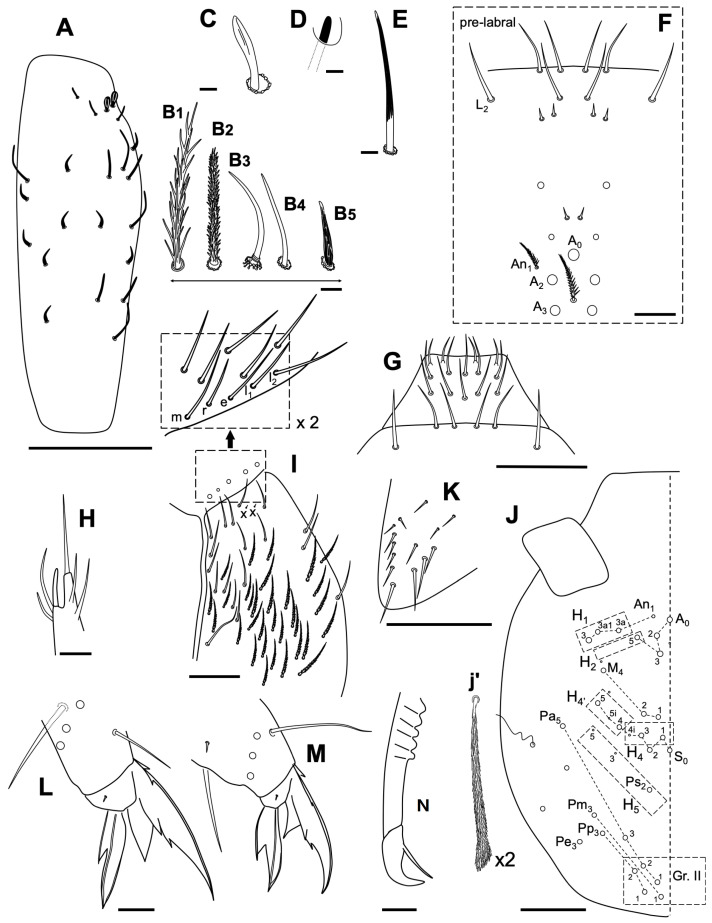
*Coecobrya octoseta* Jordana & Baquero sp. nov. (**A**) Ant III segment; (**B_1_**–**B_5_**) different chaetae and sensillae of Ant III–IV segments ((**B_1_**) ciliated chaeta, present with different lengths; (**B_2_**) ciliated sensillum; (**B_3_**) curved sensillum with a *torus* at its basis; (**B_4_**) simple sensillum; (**B_5_**) short sensillum); (**C**,**D**) chaeta next to special organite; (**E**) terminal special chaeta present at the tip of antenna; (**F**) clypeal area; (**G**) labral chaetae; (**H**) labial papilla E; (**I**) labial area; (**J**) dorsal head chaetotaxy (**J**) detail of one of the dorsal macrochaetae); (**K**) trochanteral organ; (**L**) distal tibiotarsus, claw and empodium of *C. tenebricosa*; (**M**) same of *C. octoseta* sp. nov.; (**N**) distal area of *dens* and mucro. The white circles represent ciliated chaetae, and their size is proportional to the size of the alveolus. Scale bars: (**A**,**G**,**I**–**K**) 0.05 mm; (**B**) 0.002 mm; (**C**–**E**) 0.001 mm; (**F**) 0.02 mm; (**H**,**L**–**N**) 0.01 mm.

**Figure 12 insects-14-00525-f012:**
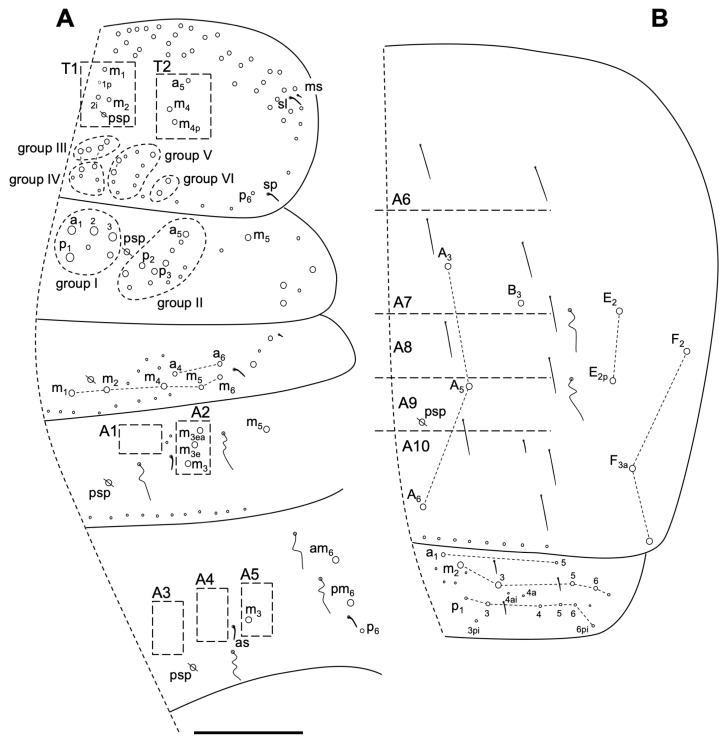
*Coecobrya octoseta* Jordana & Baquero sp. nov. dorsal macrochaetotaxy. (**A**) Th II to Abd III; (**B**) Abd IV–V. The white circles represent ciliated chaetae, and their size is proportional to the size of the alveolus. Scale bar 0.1 mm.

**Figure 13 insects-14-00525-f013:**
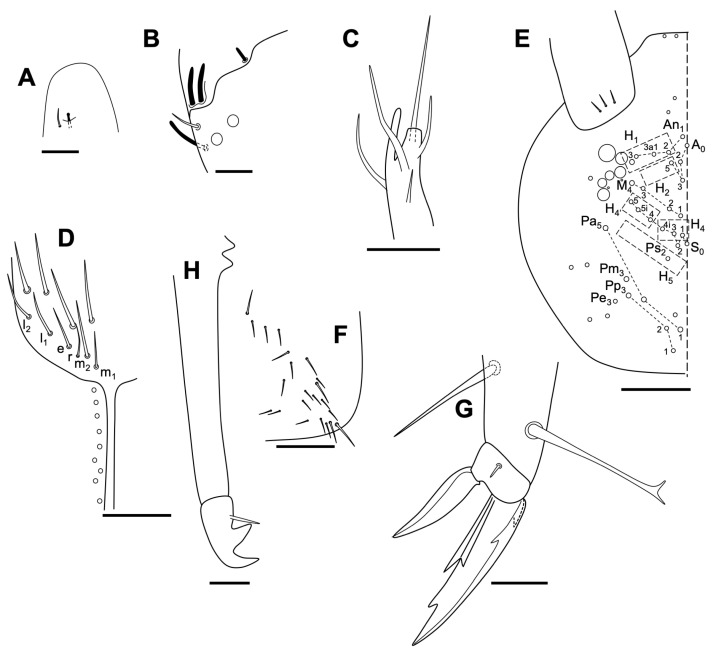
*Sinella duodecimoculata* Jordana & Baquero sp. nov. (**A**) distal area of Ant IV showing the special organite and its accompanying chaeta; (**B**) distal area of Ant III (sensory organ sensillae); (**C**) labial papilla E; (**D**) labial posterior area (posterior row and chaetae of channel); (**E**) dorsal head macrochaetotaxy; (**F**) trochanteral organ; (**G**) distal area of leg 3, claw and empodium; (**H**) distal area of *dens* and mucro. The white circles represent ciliated chaetae, and their size is proportional to the size of the alveolus. Scale bars: (**B**,**G**,**H**), 0.02 mm; (**C**,**F**) 0.05 mm; (**D**,**E**) 0.1 mm.

**Figure 14 insects-14-00525-f014:**
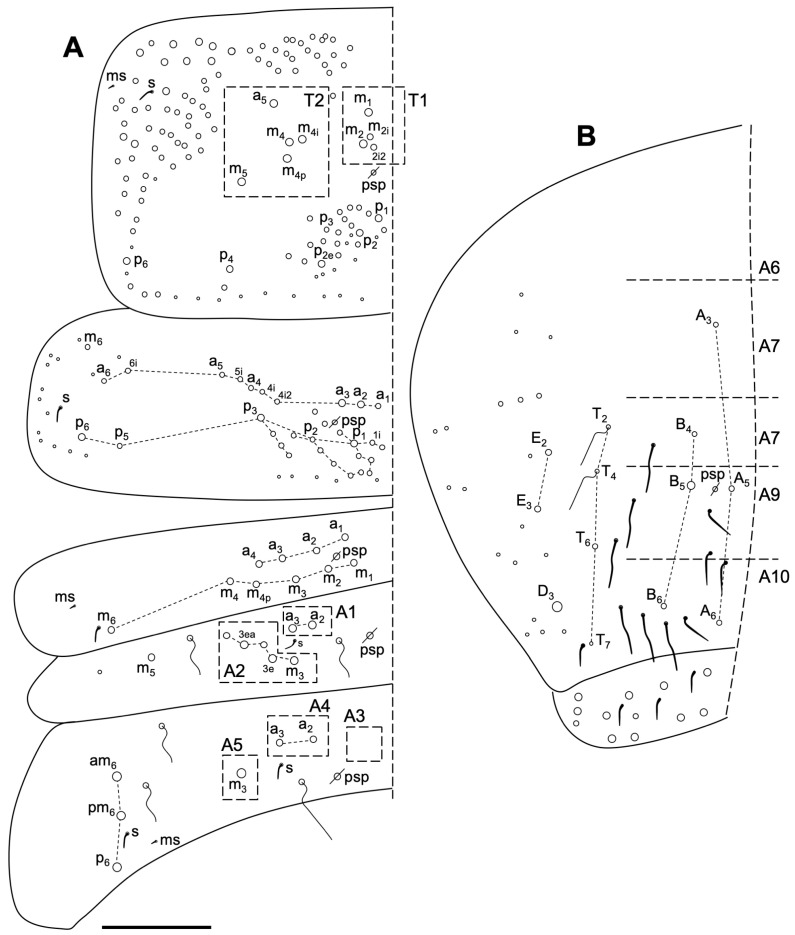
*Sinella duodecimoculata* Jordana & Baquero sp. nov. dorsal macrochaetotaxy. (**A**) Th II to Abd III; (**B**) Abd IV–V. The white circles represent ciliated chaetae, and their size is proportional to the size of the alveolus. Scale bar 0.1 mm.

**Figure 15 insects-14-00525-f015:**
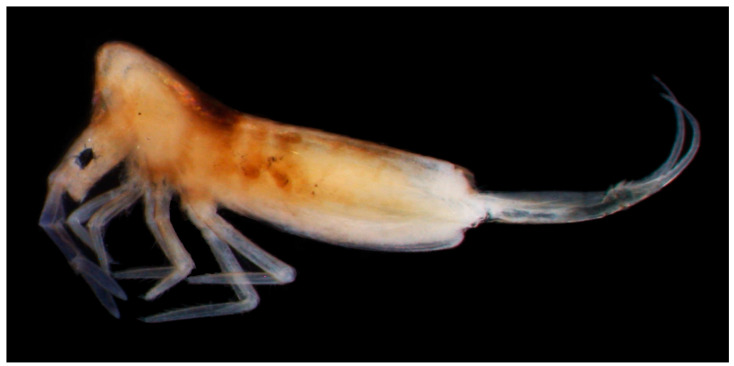
*Lepidocyrtus flexicollis* (photo: E. Baquero).

**Figure 16 insects-14-00525-f016:**
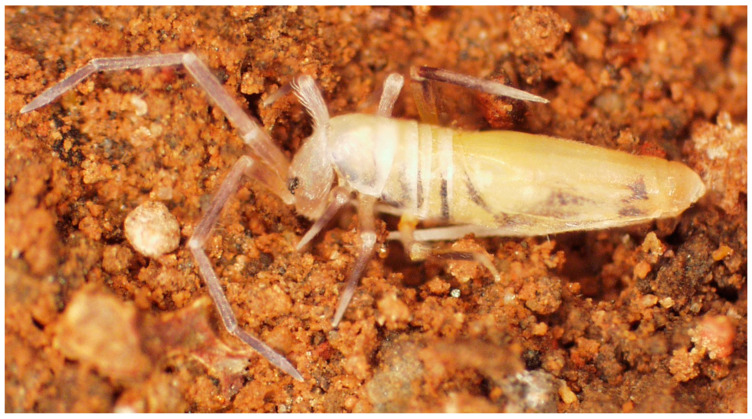
*Entomobrya pazaristei* photographed into the cave (photo: Nélio Freitas).

**Table 1 insects-14-00525-t001:** Summary table of the physical and environmental characteristics of the caves.

Cave	CVI	CVII	CVIII	LN
full extension (in meters)	300	110	92	85
maximum entrance height (in meters)	2.0	3.0	1.5	0.6
maximum entrance width (in meters)	3.5	3.0	2.0	0.6
maximum height of the tube (in meters)	10	8.0	5.0	12
minimum tube height (in meters)	0.5	0.6	0.5	0.5
relative humidity (%) *	80–95	80–95	80–95	80–95
temperature (°C)	15–18	15–17	15–95	16–17

* Values obtained in this study: CVI, Cavalum I; CVII, Cavalum II; CVIII, Cavalum III; LN, Landeiros. Other values are taken from Calandri (1991) [34].

## Data Availability

All data are contained within the article. Collected specimens have been deposited in the collection of the Museum of Zoology, Department of Environmental Biology, University of Navarra (MZNA), and the Museo Nacional de Ciencias Naturales (CSIC), Madrid (MNCN).

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
