# Peer review of "Collembola of the Cavalum and Landeiro Caves (Madeira, Portugal)†"

_insects, 2023, doi:10.3390/insects14060525_

Round 1

Reviewer 1 Report

The peer-reviewed manuscript is a comprehensive and extensive study on the Collembola fauna of Madeira's unique habitats. The subject matter fits within the scope of the journal. The results are very interesting and important from the point of view of biogeography and taxonomy. The authors are well-known specialists, which increases the reliability of the data contained in the manuscript. The work is well illustrated and written very carefully and therefore can be published without corrections.

Author Response

Thanks!

Reviewer 2 Report

The manuscript introduces a problem of conservation in Macaronesia, presents a very good taxonomic survey and species description, but it was missing the discussion of the conservation status, extinction threats and future perspectives. The survey and habitat evaluation are important contributions that can be drawn from the work. Considering that the collections were made more than 20 years ago, and the caves were open for public visitation without any control and with 150000 visitors in a single year, those species may well be extincti by now. Some public policies recomendations would be useful and improve the whole merit of the work done.

Some minor corrections and observations are marked in the revised pdf.

Author Response

Line 49. “the sampling is rather old, if the species are endangered and the cave is not controled, they may be exctinct by now”.

ANSWER. There are two cave complexes in Madeira, the one studied by us is the Cavalum complex which has not been explored for tourism or other purposes. The caves are located on private property in the middle of a small forest and no intervention has been done to alter their structure or entrance. They are open and can be visited, but are not easy to find. Their size is small and is not suitable for tourist exploitation, so visits are sporadic. For all these reasons, we believe that the species did not become extinct. Undoubtedly, more studies are needed to determine the current conservation status of the cave fauna.

Line 69. “ration”.

ANSWER. It is ok, it is “radiation”; changed.

Line 74. “this is, at least, imprecise, many non-human size access underground spaces are suitable for subterranean fauna”.

ANSWER. We agree, we have discussed it ourselves many times. In the volcanic tubes there are also fissures. We think that the sentence can be fixed by changing "to live" to "to prospect".

Line 133. “20+ years with 150000 visitors in a single year. It is urgent to visit and protect the place”

ANSWER. Caves with more than 150 000 visitors in a single year are from a different volcanic complex (Sain Vicente) not to Cavalum.

Line 201. “there?”

ANSWER. There, is true; changed.

Line 229. Highlighted text.

ANSWER. Part of the text of the sentence was missing. Now it is completed.

Line 340. “I believe that type locality must be a single place, the other localities must be additional records.

ANSWER. We think not, since specimens from the three caves have been used for the description, and they are paratypes, then they belong to the type locality, distant between them about 500 m. Therefore, we consider that the three caves describe the type locality. Then we put the municipality, the name of the island, and the country, which allows us to locate the capture area of the specimens perfectly. The three caves share the same faunal complex, that indicates they are the same locality.

Line 341. “this is the type locality”

ANSWER. Yes, it is part of the Type locality. We think that in this way the place of capture of the Holotype is perfectly located. The rest has been explained in the previous comment.

Line 414. “among the remaining what?

ANSWER. The sentence has been completed, and the redaction slightly changed for aclaration.

Line 452. “se previous comment”

ANSWER. See the previous answer.

Line 454. “type locality”

ANSWER. See the previous answer.

Line 455. “Please check the recommendations, I believe even the paratypes must be from the type locality”

ANSWER. See previous answers.

Line 495 (Figure 8). “the labeling of the chaetotaxy is invisible even at 140%, please change the font size”

ANSWER. Done.

Line 517. “change and standardize font size”

ANSWER. Done.

Line 538–539. Highlighted text.

ANSWER. See previous answers.

Line 558. “flip image vertically”

ANSWER. Done.

Line 573 (Figure 11). “F, I, J. increase font size”

ANSWER. Done.

Line 601 (Figure 12). “font size”

ANSWER. Done.

Line 659. “after”

ANSWER. Done.

Line 671 (Figure 13). “font size”

ANSWER. Done.

Line 692 (Figure 14). “font size”

ANSWER. Done.

Line 784. “female subanal appendage?”

ANSWER. Yes. Changed.

Line 824. “It was missing the discussion of the conservation status, extinction threats and future perspectives. Considering that the collections were made more than 20 years ago, and the caves were open for public visitation without any control and with 150000 visitors in a single year, those species may well be extincti by now. Some public policies recomendations would be useful and improve the whole merit of the work done”

ANSWER. A paragraph has been added that we think makes this point clear.